# Enzymes of Fibrosis in Chronic Liver Disease

**DOI:** 10.3390/biomedicines10123179

**Published:** 2022-12-08

**Authors:** Ioannis Tsomidis, George Notas, Costas Xidakis, Argyro Voumvouraki, Dimitrios N Samonakis, Mairi Koulentaki, Elias Kouroumalis

**Affiliations:** 1First Department of Internal Medicine, AHEPA University Hospital, 54621 Thessaloniki, Chalkidiki, Greece; 2Laboratory of Gastroenterology and Hepatology, School of Medicine, University of Crete, 71500 Heraklion, Crete, Greece; 3Laboratory of Experimental Endocrinology, School of Medicine, University of Crete, 71500 Heraklion, Crete, Greece; 4Department of Gastroenterology, PAGNI University Hospital, School of Medicine, University of Crete, 71500 Heraklion, Crete, Greece

**Keywords:** primary biliary cholangitis, prolyl hydroxylase, matrix metalloproteinases, collagen turnover

## Abstract

Introduction: Liver fibrosis has been extensively studied at the cellular and molecular level, but very few data exist on the final enzymatic stages of collagen synthesis (prolyl hydroxylase, PH) and degradation (matrix metalloproteinases, MMPs), particularly in primary biliary cholangitis (PBC). Aim: We studied enzyme activities in liver tissue from patients with chronic liver diseases and compared them to normal livers. Patients: Eighteen patients with PBC of early and late stages (Ludwig’s classification) and seven on treatment with ursodeoxycholate (UDCA) were studied and compared to 34 patients with alcoholic liver disease (ALD), 25 patients with chronic viral liver disease and five normal biopsies. Sera were available from a total of 140 patients. Methods: The tritiated water released from the tritiated proline was measured in PH assessment. ^14^C intact and heat-denatured collagen substrates were used to measure collagenase and gelatinases, respectively. ^3^H Elastin was the substrate for elastase. In serum, ELISAs were used for MMP-1, TIMP-1, and TIMP-2 measurements while MMP-2 and MMP-9 were estimated by zymography. Results: PH was significantly increased in early and late PBC. Collagenase was reduced only in the late stages (*p* < 0.01), where the ratio PH/collagenase was increased. UDCA treatment restored values to almost normal. Gelatinases were reduced in late stages (*p* < 0.05). In contrast to PBC and ALD fibrosis, collagen synthesis is not increased in viral fibrosis. The balance shifted towards collagen deposition due to reduced degradation. Interestingly, gelatinolytic activity is not impaired in ALD. Elastase was similar to controls in all diseases studied. TIMP-1 was reduced in early PBC and viral and alcoholic hepatitis and cirrhosis (*p* < 0.001). Conclusions: (1) There is evidence that collagen synthesis increases in the early stages of PBC, but the collagenolytic mechanism may compensate for the increased synthesis. (2) In viral disease, fibrosis may be due to decreased degradation rather than increased synthesis. (3) The final biochemical stages of liver fibrosis may be quantitatively different according to underlying etiology.

## 1. Introduction

Liver fibrogenesis is a normal phenomenon of tissue repair. It is regulated by a network of interrelated signaling interactions between the liver parenchymal cells (hepatocytes), non-parenchymal cells, and various infiltrating immune cells [1,2].

Liver fibrosis has been extensively studied in many experimental animals, and the underlying molecular mechanisms have been elucidated. Hepatic stellate cells (HSCs) produce the extracellular matrix involved in fibrosis. Kupffer cells (KCs) are also implicated through the production of transforming growth factor-β (TGF-β), the primary pro-fibrotic signal to HSCs, but also through the production of matrix metalloproteinases (MMPs) that regulate the degradation of collagen and other constituents of extracellular matrix (ECM) [3,4].

ECM was initially considered a simple scaffold for liver cells. Now, it is recognized as a dynamic factor in development and disease with a role in regulating cell behavior [5].

The initial definition of the ECM as a segregation of fibrillary proteins, such as collagens, glycoproteins, and proteoglycans, has been broadened to include ECM-affiliated proteins and ECM modifiers such as lysyl oxidases, MMPs and factors bound to the ECM such as TGF-β, and various cytokines. Advanced fibrosis is also associated with dense ECM, rich in elastin, which has been used as a marker for the chronicity of fibrosis [6]. Extracellular matrix deposition is both the result and a driver of fibrosis [7].

The fundamental driver of fibrosis is the damage of hepatocytes and the resultant sterile inflammation, which results from the activation of several inflammasomes by pathogen-associated molecular patterns (PAMPS) and damage-associated molecular patterns (DAMPS), not initiated by microbes. However, different exogenous and endogenous factors regulate the process of fibrosis. Autophagy is an essential regulator of fibrosis [8]. Serum exosomes are another regulator. They may be either profibrotic activating HSCs through the miR-574-5p content [9], or anti-fibrotic like those secreted by natural killer cells [10]. Single-cell RNA sequencing revealed the role of individual cells in fibrosis and led to the concept of fibrotic niche and the transcriptomes of individual cells [11,12].

The resolution of hepatic fibrosis is mediated not only by a decrease in the production of ECM but also by an increase in the degradation of existing ECM by matrix metalloproteinases (MMPs) [13,14].

Early reports have indicated that hepatic prolyl hydroxylase is closely related to the rate of collagen synthesis, even though it is not the rate-limiting synthetic enzyme [15,16,17].

On the other hand, MMPs and their inhibitors are involved in fibrosis resolution degrading the components of ECM. Twenty-four different MMPs have been discovered in humans. They are categorized into six classes: (i) collagenases, (ii) stromelysins, (iii) gelatinases, (iv) matrilysins, (v) membrane-type MMPs, and (vi) others. MMPs are secreted as proenzymes and activated in extracellular space [18,19]. MMP-1 (EC 3.4.24.7), is the main enzyme degrading fibrillary collagen. MMP-2 (gelatinase A; EC 3.4.24.24, 72-kDa) and MMP-9 (gelatinase B; EC 3.4.24.35, 92-kDa) are the two main components of the space of Disse, and they are involved in the degradation of collagen IV, fibronectin, and elastin. The initial cleavage of fibrillar collagens by MMP-1 or MMP-13 renders them susceptible to degradation to small fragments by gelatinases. The main MMP-degrading elastin is MMP-12 or metalloelastase [6,20,21].

HSCs and Kupffer cells are the primary producers of metalloproteinases [22]. MMP-2 and MMP-9 are also expressed in different subsets of leukocytes, such as T cells, neutrophils, monocytes, and liver-infiltrating macrophages [23,24].

MMPs may be differentially involved in the fibrotic process of viral and non-viral chronic liver diseases. There are differences between HBV and HCV chronic hepatitis and early and late fibrosis [25]. Therefore, it is unclear if the final steps of fibrosis are identical irrespective of etiology, as alternative pathways exist [26,27]. Information on the final biochemical stages of liver fibrosis is few for individual diseases, particularly for primary biliary cholangitis (PBC).

PBC is a chronic cholestatic liver disease characterized by the destruction of small to medium size intrahepatic bile ducts with obscure pathogenesis. An interplay of genetic, epigenetic, and environmental factors is implicated. Apoptosis and senescence of biliary epithelial cells (BECs) are fundamental pathogenetic factors accompanied by damage in the biliary bicarbonate umbrella that protects epithelial cells from the toxic effects of bile acids. The integrity of the bicarbonate umbrella is maintained by the chloride/bicarbonate anion exchanger 2 (AE2). A dysregulation of AE2 makes BECs vulnerable to apoptosis and senescence, which may be the initial trigger for activation of the innate and adaptive immunity leading to further damage and cholestasis. Kupffer cells engulf senescent and apoptotic cells, and when their autophagic capacity is overwhelmed, they secrete mediators such as TNFa, IL-1, and TGFβ that perpetuate inflammation and activate hepatic stellate cells promoting fibrosis. Ursodeoxycholic acid is still the first-line treatment for PBC, followed by obeticholic acid, a highly selective agonist of the farnesoid X receptor. Peroxisome proliferator-activated receptor (PPAR) agonists such as fenofibrate (PPARα agonist), bezafibrate (a pan α, β/δ, γ PPAR agonist), elafibranor, (a PPARα and PPARδ agonist), and seladelpar (a selective PPAR δ agonist) are undergoing clinical trials [28,29,30].

Moreover, simultaneous collagen synthesis and degradation in different liver diseases have not been reported. Therefore, the purpose of the present study was to measure, at the same time, both the synthesis and degradation of collagen in liver tissue in PBC and other chronic liver diseases. A second purpose was to identify possible differences in the biochemical mechanisms of fibrosis in different chronic liver diseases. We studied the enzymatic activities of the main enzymes involved in the synthesis and degradation of extracellular matrix in liver tissue. We selected PBC, a model of cholestatic immune-mediated liver disease that lacks published information on fibrosis turnover. Alcoholic liver disease (ALD), one of the most common liver diseases in the western world, and chronic viral liver disease were chosen as representatives of diseases with entirely different pathophysiology. PH activity has been used as a measure of collagen synthesis. Metalloproteases central to collagen degradation, such as collagenase (MMP-1) and gelatinases (MMP-2, MMP-9), were also measured to assess fibrosis resolution. Elastase, an enzyme degrading elastin, was also measured. Serum levels of MMPs and tissue inhibitors of metalloproteinases (TIMP-1 and TIMP-2) were additionally assessed.

## 2. Material and Methods

### 2.1. Patients

Diagnosis in all patients was verified by liver biopsy. All cirrhotic patients had compensated cirrhosis. All patients provided written informed consent, and the research protocol was approved by the Ethics Committee of the University Hospital (3468/B/2022 and 9113/2022). This study was carried out in accordance with the principles expressed in the Declaration of Helsinki.

Liver biopsies from 25 patients (23 females, age 35–60) with PBC, seven with early disease stages (I or II), and eleven with late disease stages (III or IV) according to Ludwig’s classification [31], as well as seven on treatment with UDCA (stage III or IV) for at least one year, were studied. The diagnosis of PBC was established using generally accepted criteria, according to the European Association for the Study of the Liver (EASL) guidelines for PBC [32,33]. All PBC patients were AMA positive by immunofluorescence, anti-M2 positive by ELISA, and had increased IgM levels. They were also sp100 and gp210 positive.

Thirty-four patients with ALD (20 males, age 25–57 years, 14 with fatty liver, 8 with acute alcoholic hepatitis, 12 with cirrhosis) and 25 patients with chronic viral liver disease ((17 males, age 28–61, 16 with HBV, nine with HCV), 13 with chronic moderate or severe hepatitis, 12 with cirrhosis), were also studied.

Daily alcohol consumption ≥40 g and ≥20 g of ethanol for men and women, respectively, for at least five years and the absence of other etiologies of chronic liver disease, particularly of obesity and diabetes, were the criteria for diagnosis of ALD.

Five normal liver biopsies (tested for suspected liver tumors) were used as normal controls.

### 2.2. Material and Methods

Ferrous ammonium sulfate from British Drug Houses, Liverpool, England; Bovine elastin and all other reagents from Sigma, London, England; 3,4(n)-^3^H proline and U-^14^C glycine were purchased from Perkin Elmer. Tritiated NaBH_4_ was from Rc Tritec, Switzerland.

Patients from whom liver biopsies were obtained for diagnostic purposes served as controls. Approximately one-third of the tissue obtained was separated for enzyme determination, and the remainder was fixed in 10% formaldehyde for histological examination.

The specimens were quickly frozen within 90 sec in liquid nitrogen and then stored at −40 °C for a maximum period of 5 days before enzyme assays were performed.

#### 2.2.1. Prolyl Hydroxylase

The samples for the enzyme determination were homogenized at 4 °C in a glass motor-driven homogenizer with 0.5 mL of 0.1 M Tris/HCl buffer, pH 7.5 containing 0.25 M sucrose, 10^−5^ M ethylenediaminetetraacetic acid, 10^−3^ M dithiothreitol, 0.1% Triton X-100, and 50 μg/mL of phenylmethylsulphone fluoride.

The ^3^H proline-labeled substrate was prepared by incubating 6 g of minced chick embryo in 8.5 mL Krebs-Ringer buffer with 5mCi 3,4-^3^H-labeled proline in the presence of 1 mM a+‘-dipyridyl as described by Hutton et al. [34]. The substrate was extracted into cold 0.5 N acetic acid and extensively dialyzed. Ten ml aliquots of the substrate were stored at −20 °C and thawed immediately before use. The substrate solutions contained 2 mg of protein per ml with specific activity varying between 75,000 and 200,000 dpm per mg of protein.

Collagen proline hydroxylase activity was assayed by the method of Hutton et al. [35], as described by Mezey et al. [36] and Mc Gee et al. [37] for liver biopsy samples, using the modified cofactor concentrations and standardization of the assay introduced by Stein et al. [38].

This method measures the tritiated water formed when the tritiated proline in the collagen substrate is hydroxylated to hydroxyproline. The reaction mixture consisted of ^3^H proline-labeled substrate, 0.1 mL; tissue homogenate, 0.1 mL; a-ketoglutarate, 1 mM; ascorbic acid, 5 mM; ferrous ammonium sulfate, 1 mM; all in a volume of 1 mL of 0.1 M Tris/HCl buffer, pH 7.5. The reaction mixture was incubated aerobically for 30 min at 30 °C, and the reaction was stopped by adding 0.2 mL of 50% trichloroacetic acid (TCA). The tritiated water produced was separated by vacuum distillation and counted in Bray’s solution in a Packard Tri-Carb scintillation counter, model 3375. The reaction was linear up to 30 min with liver tissue between 5 and 30 mg wet weight. Enzyme activities were expressed as dpm/mg Protein/30 min.

#### 2.2.2. Collagenase

Liver tissue was homogenized in 1 mL of 50 mM tris HCl, pH 7, with 0.2 M NaC1, 5 mM CaCI, and 0.1% Triton X-100.

^14^C-labelled guinea pig skin collagen was used as the substrate [39]. Guinea pig skin collagen was labeled in vivo with ^1^⁴C-glycine and extracted with 1 M NaCl and 0.5 M acetic acid. After extensive dialysis against 0.02 M disodium orthophosphate and adjustment to give concentrations of 2.69 mg collagen/mL, the substrates were lyophilized. After reconstitution, one volume of the neutral salt extract (17.000 dpm/mg) was added to 2 volumes of the acid extract (3.500 dpm/mg). A quantity of 200 μL of the mixture was added to each tube (approximately 4.400 dpm).

The dioxane precipitation method was employed [40,41] with chloromercuribenzoic acid activation of the latent enzyme [42].

The incubation mixture (0.4 mL final volume) contained a final concentration of 7 mM CaCl₂, 0.25 M glucose, and 0.15 M NaCl. Incubation was carried out at 37 °C for 120 min. A quantity of 20 μL of 80 mM o-phenanthroline was added, and the mixture was kept at 35 °C for 60 min before returning to room temperature; then, 0.4 mL of dioxane was added, and the tubes were centrifuged in a Beckman microfuge for 4 min. A quantity 200 μL of the supernatant was added to 9 mL scintillation fluid (cocktail T, BDH) and counted in a Hewlett Packard liquid scintillation spectrometer. To activate latent collagenase, p-chloromercuribenzoic acid was added to the homogenate 30 min before the assay for a concentration of 3 mM; a similar concentration of PCMB was included in the incubation mixture.

A quantity of 10 μg of trypsin, instead of homogenate, was added to 2 tubes in each run to check the suitability of the substrate, which was discarded if more than 5% of the total counts were released. Collagenase activity was expressed as dpm per mg protein per hour.

For blanks, samples were incubated in the presence of 10mM EDTA, a potent inhibitor of collagenase [43].

The liver biopsy was repeatedly washed with 0.9% NaCl to remove contaminated serum, which might inhibit collagenase activity (7), and then blotted, weighed, and kept at −20 °C.

#### 2.2.3. Gelatinase 

A ^14^C collagen solution, prepared as described above, was used for the gelatinase assay after heating at 45 °C for 15 min to form ^14^C-gelatin [44,45,46]. A quantity of 50 μL of the substrate (2.69 mg/mL gelatin) was added to 50 μL buffer (0.1 M Tris-HCl, pH 7.6) and 100 μL of the sample. The incubation mixture also contained 0.15 M NaCl and 5 mM CaCl₂. After incubation for 120 min at 37 °C, 100 μL of 50% (*w*/*v*) trichloroacetic acid was added, and the tube was placed on ice for 5 min and then centrifuged for 4 min in a Beckman microfuge. A quantity of 200 μL of the supernatant was counted for radioactivity in 9 mL of scintillation fluid. For the blanks, EDTA (10 mM) was included in the incubation mixture. Enzyme activity was expressed as dpm per mg protein per hour.

#### 2.2.4. Elastase

Reduced ^3^H-bovine neck ligament elastin was prepared according to Takahashi et al. [47]. Ten milligrams of Bovine elastin (Sigma) suspension in distilled water was incubated with tritiated NaBH_4_ (0.1 μCi/mg elastin). The reaction was terminated by the addition of acetic acid to pH 3.0. Elastin was removed by centrifugation, washed with distilled water, and lyophilized. A 0.5% (*w*/*v*) ^3^H-elastin suspension was prepared in 0.05 M Tris-HCl buffer pH 8.0. The incubation mixture contained 100 μL of this suspension, 100 μL of 0.05 M Tris-HCl buffer pH 8.0, and 100 μL of sample homogenate. After shaking for 60 min at 37 °C, the reaction was terminated by adding 50 μL diisopropylphosphorofluoridate (DFP) (final concentration 0.04 M). The mixture was centrifuged in a Beckman microfuge for 3 min, and 100 μL of the supernatant was counted for radioactivity. Bovine serum albumin blanks (0.05 mg/mL) were run simultaneously, and the enzyme activity was expressed as dpm per mg protein per hour.

#### 2.2.5. Protein Estimations

The method of Lowry et al. [48] was used, with bovine serum albumin as the standard.

#### 2.2.6. Serum Studies

Sera from 140 patients and 10 normal controls were available. Forty-six patients were diagnosed with PBC (18 early, 18 late, and 10 were on treatment with ursodeoxycholic acid), 17 patients had viral compensated cirrhosis (10 with HBV, seven with HCV), 26 had chronic viral hepatitis (16 with HBV, 10 with HCV), and 51 had ALD (16 fatty liver, 19 acute alcoholic hepatitis, and 16 alcoholic cirrhosis).

Blood was collected from the patients within the first 3 days after hospital admission. Serum samples were separated after blood clotting by centrifugation at 1100 g for 10 min and stored at −80 °C until measured [49].

Serum protein levels of MMP-1 (EC 3.4.24.7), MMP-2 (3.4.24.24), MMP-9 (3.4.24.35), TIMP-1, and TIMP-2 were determined by Quantikine enzyme-linked immunosorbent assays (R&D systems) according to manufacturer instructions. The MMP-1 ELISA recognizes pro-MMP, free MMP, and MMP complexed with inhibitors such as TIMP-1 but not the a2 macroglobulin-MMP-1 complexes. Coefficient of variations (%CV) were in the range of 7–11% in all ELISAs.

The serum levels of MMP-2 and MMP-9 were also measured by a zymogram protease assay, which recognizes both active, and precursor forms of circulating enzymes free or complexed with TIMPs. Briefly, 10% of sodium dodecyl sulfate (SDS)–polyacrylamide gels containing 0.1% gelatin as the substrate (Zymogram ready gels, Bio-Rad Laboratories, Hercules, CA, USA) were used for electrophoresis. After electrophoresis, the gel was washed with 1.5% Triton X-100 on a shaker for 60 min to remove SDS. The gel was then incubated overnight in reaction buffer (50 mM Tris/HCl, pH 8.0, 200 mM NaCl, 5 mM CaCl2, 0.2% Brij 35) at 37 C, stained with Coomasie brilliant blue R-250 and destained with 10% acetic acid-20% methanol in water. Quantification of MMP activity was performed using densitometry linked to a gel documentation (Gel-doc 2000) and analysis system (QuantiscanTM, Biosoft, Cambridge, UK). To normalize the possible difference between zymograms, serial dilutions of recombinant activated MMP-2 and MMP-9 (Oncogene Research Products, San Diego, CA, USA) were incorporated in every gel, and total serum MMP-2 or MMP-9 activity of samples was calculated from generated standard curves [50].

### 2.3. Statistical Analysis

Data are shown as means ± SD. Hepatic enzyme results are shown as box charts with normal distribution curve. Statistical analyses were performed using GraphPad Prism 9.3.0 and OriginPro 2018 SR1 software. Bartlett test was used to assess the variance of SDs. Statistical significance was calculated by one-way ANOVA (Welch correction in cases of unequal variances). Kruskal–Wallis test (the non-parametric one-way ANOVA) was used when the Kolmogorov–Smirnov test showed a non-canonical distribution of data. *p* < 0.05 was considered statistically significant. All differences are in comparison with the normal controls.

## 3. Results

### 3.1. Hepatic Prolyl-Hydroxylase

As shown in Figure 1, hepatic prolyl hydroxylase showed significantly increased activity in early PBC (2923 ± 817 dpm/mg Pr/30 min, *p* = 0.04) as well as in late PBC (3228 ± 822 dpm/mg Pr/30 min, *p* = 0.004) compared to normals (1760 ± 435 dpm/mg Pr/30 min). Additionally, hepatic prolyl hydroxylase showed significantly increased activity in alcoholic hepatitis (4413 ± 1235 dpm/mg Pr/30 min, *p* = 0.002) and alcoholic cirrhosis (3788 ± 1671 dpm/mg Pr/30 min, *p* = 0.05) compared to normals. On the contrary, all other groups (on UDCA (1414 ± 139 dpm/mg Pr/30 min), viral hepatitis (2520 ± 385 dpm/mg Pr/30 min), viral cirrhosis (2150 ± 562 dpm/mg Pr/30 min), and fatty liver (1939 ± 713 dpm/mg Pr/30 min)) showed no statistically significant change of enzymatic activity compared to normals.

### 3.2. Hepatic Collagenase

As shown in Figure 2, hepatic collagenase showed significantly decreased activity only in late PBC (1060 ± 304 dpm/mg Pr/hr, *p* < 0.01) and not in early PBC (1806 ± 299 dpm/mg Pr/hr) compared to normals (1780 ± 208 dpm/mg Pr/hr). Additionally, hepatic collagenase showed significantly decreased activity not only in viral hepatitis (1127 ± 283 dpm/mg Pr/hr, *p* < 0.01) and viral cirrhosis (1112 ± 167 dpm/mg Pr/hr, *p* < 0.01) but also in alcoholic hepatitis (912 ± 215 dpm/mg Pr/hr, *p* < 0.05) and alcoholic cirrhosis (825 ± 270 dpm/mg Pr/hr, *p* < 0.05) when compared to normals. On the contrary, UDCA (2051 ± 390 dpm/mg Pr/hr) and fatty liver group (1646 ± 861 dpm/mg Pr/hr) showed no statistically significant change of enzymatic activity compared to normals.

### 3.3. Prolyl Hydroxylase/Collagenase Ratio

As shown in Figure 3, the prolyl hydroxylase/collagenase ratio was significantly increased in late PBC (2.0 ± 0.45, *p* = 0.009) but not in early PBC (1.2 ± 0.27) compared to normals (1.14 ± 0.32). Additionally, there was a significant increase of the ratio not only in viral hepatitis (1.9 ± 0.39, *p* < 0.01) and viral cirrhosis (1.87 ± 0.47, *p* = 0.05) group but also in the alcoholic hepatitis (4.55 ± 1.86, *p* = 0.006) and alcoholic cirrhosis (3.21 ± 1.22, *p* < 0.01) groups. On the contrary, there was no significant change of the ratio in the UDCA (1.03 ± 0.22) and the fatty liver (1.47 ± 0.60) group compared to normals.

### 3.4. Hepatic Gelatinase

As shown in Figure 4, hepatic gelatinase showed significantly decreased activity only in late PBC (1018 ± 159 dpm/mg Pr/hr, *p* < 0.05) and viral cirrhosis (830 ± 148 dpm/mg Pr/hr, *p* < 0.001) compared to normals (1816 ± 371 dpm/mg Pr/hr). On the contrary, no significant change of enzymatic activity was observed among the rest groups (early PBC (1300 ± 445 dpm/mg Pr/hr), UDCA (1427 ± 362 dpm/mg Pr/hr), viral hepatitis (1741 ± 486 dpm/mg Pr/hr), fatty liver (2290 ± 1146 dpm/mg Pr/hr), alcoholic hepatitis (2288 ± 1104 dpm/mg Pr/hr), and alcoholic cirrhosis (2754 ± 758 dpm/mg Pr/hr) group) compared to normals.

### 3.5. Hepatic Elastase

As shown in Figure 5, there was no significant change in hepatic elastase’s activity in all studied groups (early PBC (20.89 ± 5.17 dpm/mg Pr/hr), late PBC (22.15 ± 5.34 dpm/mg Pr/hr), UDCA (20.33 ± 3.24 dpm/mg Pr/hr), viral hepatitis (20.65 ± 12.02 dpm/mg Pr/hr), viral cirrhosis (24.40 ± 3.90 dpm/mg Pr/hr), fatty liver (20.65 ± 12.09 dpm/mg Pr/hr), alcoholic hepatitis (18.69 ± 9.81 dpm/mg Pr/hr), and alcoholic cirrhosis (19.19 ± 3.22 dpm/mg Pr/hr)) compared to normals (20.90 ± 3.79 dpm/mg Pr/hr).

It should be noted that no significant sex differences were found. However, the comparisons may be misleading due to the limited number of men studied (only two in PBC, as expected).

### 3.6. Serum MMP-1

As shown in Figure 6, serum MMP-1 showed significantly decreased activity only in late PBC (62 ± 17 ng/mL, *p* = 0.05) and not in early PBC (114.17 ± 59.64 ng/mL) compared to normals (121.79 ± 57.9 ng/mL). Additionally, serum MMP-1 showed significantly decreased activity not only in viral hepatitis (58.47 ± 44.59 ng/mL, *p* < 0.05) and viral cirrhosis (52.41 ± 39.72 ng/mL, *p* = 0.03) but also in alcoholic hepatitis (61.42 ± 29.05 ng/mL, *p* = 0.05) and alcoholic cirrhosis (50.45 ± 28.14 ng/mL, *p* = 0.02) compared to normals. On the contrary, UDCA (75.8 ± 37.38 ng/mL) and fatty liver group (109.67 ± 37.99 ng/mL) showed no statistically significant change of enzymatic activity compared to normals.

### 3.7. Serum MMP-2

As shown in Figure 7, serum MMP-2 showed significantly decreased activity in early PBC (300.22 ± 78.01 ng/mL, *p* < 0.0001) as well as in late PBC (294.26 ± 60.62 ng/mL, *p* < 0.0001) compared to normals (426.47 ± 44.93 ng/mL). Additionally, serum MMP2 showed significantly decreased activity not only in viral hepatitis (313.17 ± 90.39 ng/mL, *p* = 0.0002) and viral cirrhosis (332.01 ± 47.4 ng/mL, *p* = 0.0004) but also in alcoholic hepatitis (306.41 ± 49.45 ng/mL, *p* < 0.0001) and alcoholic cirrhosis (326.43 ± 54.87 ng/mL, *p* = 0.0004) when compared to normals. On the contrary, UDCA (353.56 ± 81.18 ng/mL) and fatty liver group (385.17 ± 46.52 ng/mL) showed no statistically significant change in enzymatic activity compared to normals.

### 3.8. Serum MMP-9

As shown in Figure 8, serum MMP-9 showed significantly decreased activity in early PBC (4.31 ± 1.67 ng/mL, *p* = 0.05) as well as in late PBC (3.34 ± 1.36 ng/mL, *p* = 0.0001) compared to normals (5.65 ± 1.48 ng/mL). Additionally, serum MMP9 showed significantly decreased activity not only in viral hepatitis (2.99 ± 1.40 ng/mL, *p* < 0.0001) and viral cirrhosis (2.80 ± 1.36 ng/mL, *p* < 0.0001) but also in alcoholic hepatitis (2.84 ± 1.09 ng/mL, *p* < 0.0001) and alcoholic cirrhosis (1.32 ± 1.08 ng/mL, *p* < 0.0001) when compared to normals. On the contrary, both UDCA (5.01 ± 1.09 ng/mL) and fatty liver group (5.09 ± 0.97 ng/mL) showed no statistically significant change of enzymatic activity compared to normals.

### 3.9. Serum TIMP-1

As shown in Figure 9, serum TIMP-1 showed significantly decreased activity in early PBC (508.9 ± 123.88 ng/mL, *p* < 0.0001) but not in late PBC (855.29 ± 214.92 ng/mL) compared to normals (1023.79 ± 180.81 ng/mL). Additionally, serum TIMP1 showed significantly decreased activity not only in viral hepatitis (342.49 ± 184.53 ng/mL, *p* < 0.0001) and viral cirrhosis (485.64 ± 116.27 ng/mL, *p* < 0.0001) but also in alcoholic hepatitis (467.82 ± 129.75 ng/mL, *p* < 0.0001) and alcoholic cirrhosis (761.9 ± 241.76 ng/mL, *p* = 0.03) when compared to normals. On the contrary, UDCA (1027.22 ± 204.69 ng/mL) and fatty liver group (985.62 ± 201.12 ng/mL) showed no statistically significant change of enzymatic activity compared to normals.

### 3.10. Serum TIMP-2

As shown in Figure 10, serum TIMP-2 showed significantly increased activity in viral cirrhosis (114.78 ± 29.41 ng/mL, *p* < 0.0001) as well as in alcoholic cirrhosis (171.87 ± 31.87 ng/mL, *p* < 0.0001) compared to normals (64.12 ± 23.61 ng/mL). On the contrary, no significant change of the enzymatic activity was observed among the rest groups (early PBC (57.9 ± 15.38), late PBC (78.41 ± 27.21), UDCA (68.76 ± 19.51), viral hepatitis (60.01 ± 25.15), alcoholic hepatitis (56.96 ± 27.92), and fatty liver (62.12 ± 20.11)) compared to normals.

## 4. Discussion

Advanced fibrosis and cirrhosis are the end stages of many chronic liver diseases indicating the existence of a common pathway in diseases of diverse etiology, such as ALD, viral hepatitis, and PBC. Irrespective of etiology or the multitude of molecular mechanisms leading to cirrhosis, the outcome depends on the balance between the production and degradation of the extracellular matrix [27]. The balance between MMPs/TIMPs regulates ECM turnover and remodeling during normal development and pathogenesis [51,52].

Proteolytic enzymes that degrade the ECM, including gelatinases (MMP-2, MMP-9) and collagenases (MMP-1 and MMP-13), are mostly produced by Kupffer cells and other liver [53,54].

This complexity of cellular origin and the extensive modifications of MMPs was the reason we chose to measure enzyme activities, as they probably represent better the facts leading to fibrosis. This is particularly true for the assay we used in collagenase estimation in liver tissue. Its main advantage is that it measures the activity of enzymes that cleave triple helical collagen distinguishing this activity from other proteinases. The total collagenolytic activity cannot discriminate between MMP1 and other enzymes that also cleave triple helical collagen (MMP-2 and MT1-MMP). The assay also measures the balance between collagenolytic enzymes and the level of the TIMPs and other inhibitors present in the sample, indicating the overall potential for collagen breakdown [55].

On the other hand, immunological assays such as ELISAs can accurately measure the amount of an individual MMP present but often do not distinguish between proenzyme, active enzyme, or inhibitor-complexed enzyme [56,57].

We also chose to use the ratio prolyl hydroxylase/collagenase as an indicator of the balance between synthesis and degradation of collagen instead of the commonly used MMP/TIMP ratio, as this ratio may not reflect the behavior of MMPs and TIMPs when evaluated separately [58,59] as in our study.

Clinical and experimental data on fibrosis-related enzyme activities are few and poorly documented in patients with PBC, hence we focused on our findings in this disease.

Prolyl hydroxylase activity was significantly elevated in both early and late PBC. Collagenase activity was similar to controls in the early stages of the disease but reduced considerably in the late stages. The same was observed in serum, where the method detects both free and TIMP-associated MMP-1 but not the enzyme associated with a-2 macroglobulin. This is also reflected by another indicator of collagen turnover, which is the ratio of prolyl hydroxylase over collagenase. It seems that at the early stages of PBC, the collagenolytic system counteracts the increased collagen synthesis, which is not the case in the late stages, and collagen deposition increases. The high levels found in early histological stages may indicate that the stimulus to collagen formation appears before fibrosis becomes histologically extensive. Interestingly, patients treated with ursodeoxycholic acid had values of prolyl hydroxylase and collagenase similar to controls. This finding requires further validation with a larger number of treated patients.

The explanation for our observations can only be speculative. In cholestasis, macrophage functions are affected due to the increased levels and altered composition of bile acids [60,61].

In our study, gelatinase activity and the activities of MMP-2 and MMP-9 were reduced in the livers of early and late PBC patients. Importantly, ursodeoxycholic acid treatment restored enzyme values to almost normal values. Although the number of treated patients was small, this finding may be significant, and further clinical studies are required.

In an early study, proline hydroxylase was elevated in ten out of fourteen patients with PBC but to a lesser degree than in patients with active cirrhosis [62]. Approximately 60% of PBC patients had increased MMP-3 in serum measured by ELISA, and higher values were found in patients with advanced disease (Ludwig stage III-IV) [63]. A different research approach showed results consistent with our findings. In a recent study, ECM turnover was increased in PBC. The synthesis of collagens III and IV was increased, a finding compatible with our results. Importantly, all ECM markers were inversely related to ursodeoxycholic acid treatment, in agreement with our results [64].

TIMP-1 and TIMP-2 expression were significantly increased in a small number of PBC cirrhosis. It was combined with increased MMP-2 mRNA expression, while MMP-1 expression was unaltered [65]. This differs from our findings, where TIMP-2 levels are similar to normal controls, and TIMP-1 is significantly reduced in early PBC, thus favoring collagenolysis. In contrast, liver collagenase and total gelatinase activities were reduced in late PBC. However, their conclusions are similar to our study as the ratio of prolyl hydroxylase/collagenase is significantly increased, indicating collagen deposition in advanced PBC.

There is experimental evidence that the reduction of MMP-2 and MMP-9, as observed in our study, may benefit PBC. The inhibition of MMP-2 and MMP-9 activity in intrahepatic cholestasis of pregnancy ameliorated the biliary disease [66].

There is evidence that alterations leading to ECM remodeling may differ between PBC and other liver diseases, either autoimmune- or virally-related ones [64]. A study of TGF-β isoforms has shown that the TGF-β1 isoform was significantly decreased in all cirrhotic patients compared to controls. At the same time, TGF-β2 was increased in HCV-related cirrhosis but not in PBC. TGF-β3 was characteristically increased in early and late PBC and decreased in viral cirrhosis [67]. Our findings in ALD and chronic viral disease support this idea. Patients with alcoholic hepatitis and alcoholic cirrhosis had the highest prolyl hydroxylase values, while patients with fatty liver had normal values. Earlier papers reported similar results [16,36,68]. Interestingly, there was no significant difference in patients with either chronic viral hepatitis or viral cirrhosis compared to controls.

We found a significant reduction of liver collagenase activity in ALD and chronic viral disease. Again, patients with fatty liver had levels comparable to the controls. Serum MMP-1 showed a similar picture in disagreement with a previous report where no change of serum MMP-1 was noted [69].

As expected, the prolyl hydroxylase/collagenase ratio increased in all groups except in patients with fatty liver, with the highest values found in alcoholic hepatitis and alcoholic cirrhosis. Despite normal prolyl hydroxylase values in chronic viral hepatitis, the ratio of prolyl hydroxylase/collagenase was increased, indicating collagen deposition. This is in accordance with earlier studies. Normal prolyl hydroxylase and decreased collagenase have also been reported in a carbon tetrachloride-induced cirrhosis model [70].

Total gelatinase activity was reduced in the liver of patients with viral cirrhosis but not in alcoholic hepatitis or alcoholic cirrhosis. However, MMP-2 and MMP-9 were reduced in the serum of patients with chronic viral disease and ALD. It should be noted that, unlike chronic alcoholic disease, acute ethanol intoxication increases MMP-9 and TIMP-1 [71]. It should also be noted that previous studies have reported contradictory findings on the association of serum MMP-2 and liver fibrosis [72,73]. Therefore, we believe that total liver gelatinase activity is a better indicator of the association between gelatinases and liver fibrosis resolution.

Elastin is a minor ECM component in the normal liver, but it is actively synthesized by HSCs and other fibroblasts in the fibrotic liver [74]. Several MMPs degrade elastin, including MMP-2, MMP-9, MMP-7, and MMP-10 but the main elastolytic enzyme is macrophage-derived MMP-12 [75,76]. In contrast to other MMPs, elastase activity in the liver was similar to normal controls in all patient groups. Our results possibly represent the collective action of all MMPs involved in elastin degradation. Nonetheless, the elastolytic activity was not impaired in any group of patients.

Explanations for the reduced levels of MMPs can only be speculative. A strong possibility is that low levels may indicate that Kupffer cells are not functioning properly in chronic liver disease. This has been shown in animal experiments where the production of MMP-9 during fibrosis resolution was significantly reduced after the depletion of Kupffer cells [77]. In addition, dysregulation of regulatory T cells may affect gelatinases and TIMPs [78], but data on humans do not exist.

Serum TIMP-1 was reduced in both viral and alcoholic cirrhosis and hepatitis. On the contrary, TIMP-2 was significantly decreased only in cirrhotic patients. Plasma levels of TIMP-2 are related to the stage of liver fibrosis [79]. Serum TIMP-2 levels, however, are not specific. High values of TIMP-2 were reported in patients with aortic stenosis and fibrosis-related diseases [80,81].

Patients with chronic viral hepatitis display their own distinct pattern. In a previous study, we reported that MMP-2 and MMP-9 mRNA were differently expressed in non-cirrhotic HBV and HCV chronic hepatitis patients, although no significant differences were found between them [25]. The same was true when the enzyme activities of the present study were compared between the two groups. Therefore, we grouped HBV and HCV patients in the final analysis. Previous studies report MMPs and TIMPs levels in the serum of either HCV or HBV patients separately.

It is of note that in HCV, existing results are conflicting. MMP-1 was reported to be reduced in HCV in a recent study in agreement with our results [82], and a similar MMP-1 reduction was also reported in a cohort study of combined HCV and HBV patients [83].

MMP-1 and MMP-9 were found to be decreased in METAVIR stages III and IV, while MMP-2 and TIMP-1 were increased, a finding in agreement with our results [84]. Serum MMP-9 was negatively associated with the severity of liver fibrosis in chronic HCV patients [85]. MMP-1 levels were significantly lower than controls in chronic HBV patients in accordance with our results [86]. In a study using the Ishak histological classification, as we did, a gradual decrease of MMP-9 and a gradual increase of TIMP-1 were reported in agreement with our findings [87].

In disagreement with our study, serum MMP-2 and MMP-9 and TIMPs in HCV patients were reported either similar to control levels [88] or increased [89,90,91], particularly in those with HCV/HIV co-infection and chronic HBV [92]. However, TIMP-2 was increased with advanced fibrosis in accordance with our study, where TIMP-2 was increased in both viral- and alcohol-related cirrhosis (Table 1) [93]. The reason for the discrepancies is not apparent. Variable methodologies may account for some differences. Different patient selection and classification could also be an important factor. MMP-2 and MMP-9 have been associated with inflammation rather than the degree of fibrosis [94]. It should be noted that we used the Ishak classification that incorporates both fibrosis and inflammation indicators. This is a significant difference from previous studies that classify patients according to the METAVIR system based only on fibrosis. A comparative study found variations in necro-inflammatory features between the two systems [95]. As mentioned earlier, we found no significant difference between HBV and HCV chronic hepatitis.

A recent study reported that serum concentrations of MMPs are increased in chronic HCV infection, but gelatinases are inactive during fibrosis progression. When the collagenolytic activity was evaluated, lower values were found in METAVIR stages F2, F3, and F4 compared to stage F1 [96].

Our study has several limitations. First, our approach cannot identify a particular enzyme’s cellular origin (or origins). Second, it is a small cross-sectional study with an even smaller number of liver samples. However, our approach has the advantage of identifying the end result of enzymatic activity in the liver, thus overcoming the multifactorial modifications and the lack of substrate specificity of MMPs. Moreover, direct comparisons of liver collagenase and serum values of MMP1 cannot be safe as the ELISA used detects pro-MMP1 and inhibitor-complexed MMP1 in addition to the active forms of the enzyme. However, there is a general agreement between serum and liver findings in all diseases studied.

In conclusion, our findings indicate that the balance between collagen synthesis and degradation favors collagen deposition, irrespective of the etiology of liver diseases. We found that synthesis is increased, even in the early stages of PBC, but the collagenolytic system, possibly, counteracts the increased synthesis. Collagen degradation may not compensate for the collagen synthesis at the late stages of PBC. Treatment of PBC with ursodeoxycholic acid may, in part, restore the balance, indicating that a prospective long-term study is advisable. The newly developed drugs for treating PBC should also be tested for possible effects on the enzymic pathways of fibrosis. A similar mechanism was found in alcohol-related disease but not at the fatty liver stage. Interestingly, collagen synthesis is not significantly increased in chronic viral disease compared to normal liver, but the deranged balance is due to a significant decrease in the collagenolytic mechanism. Notably, TIMP-1 is significantly reduced in viral and alcoholic hepatitis and alcoholic cirrhosis, possibly acting as a compensatory mechanism to facilitate ECM degradation.

## Figures and Tables

**Figure 1 biomedicines-10-03179-f001:**
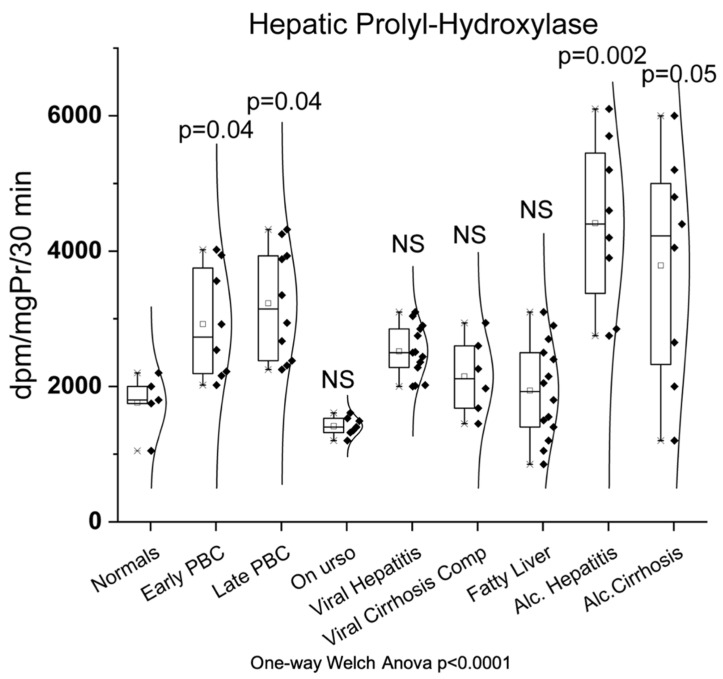
Enzymatic activity of hepatic prolyl hydroxylase in all studied groups. Box chart with points indicating the minimum, first quartile, median, third quartile, and maximum.

**Figure 2 biomedicines-10-03179-f002:**
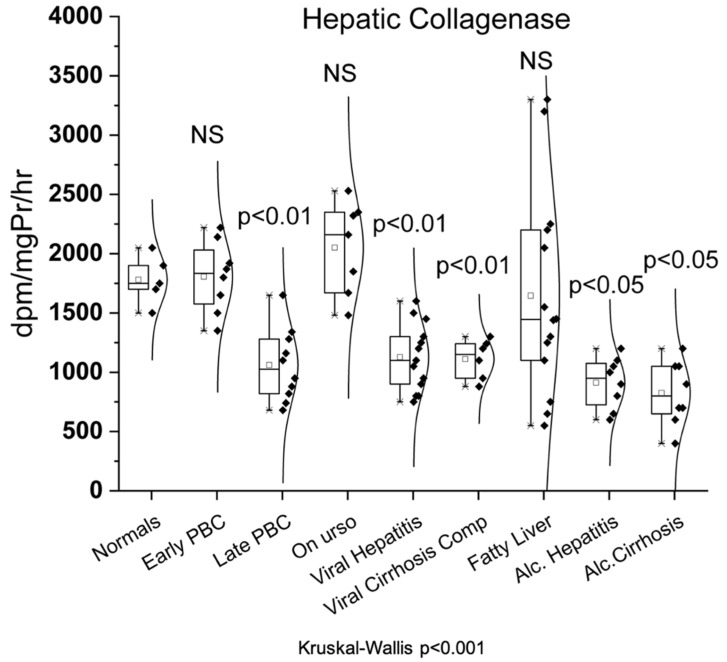
Enzymatic activity of hepatic collagenase in all studied groups.

**Figure 3 biomedicines-10-03179-f003:**
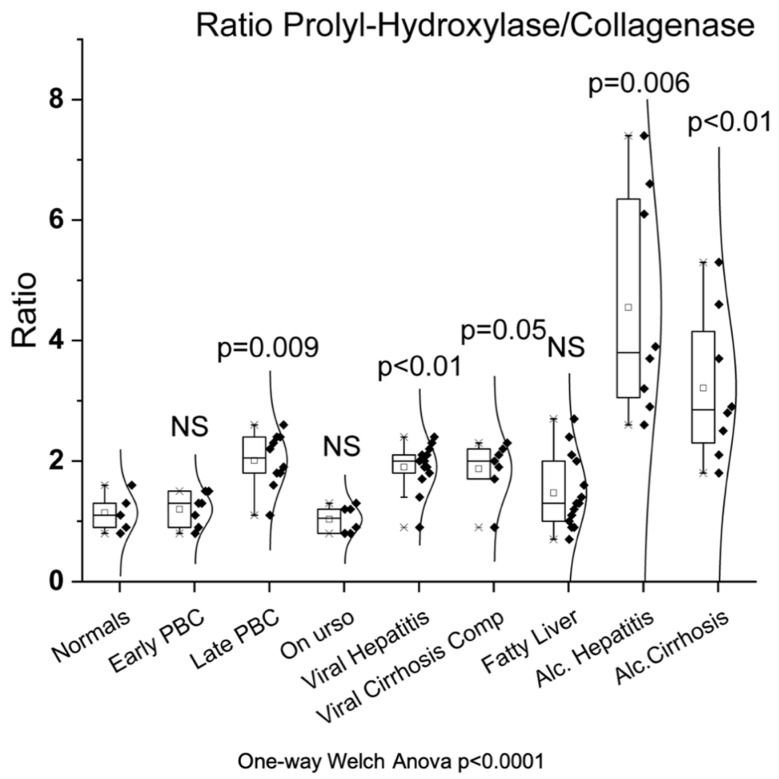
Prolyl hydroxylase/collagenase ratio in all studied groups.

**Figure 4 biomedicines-10-03179-f004:**
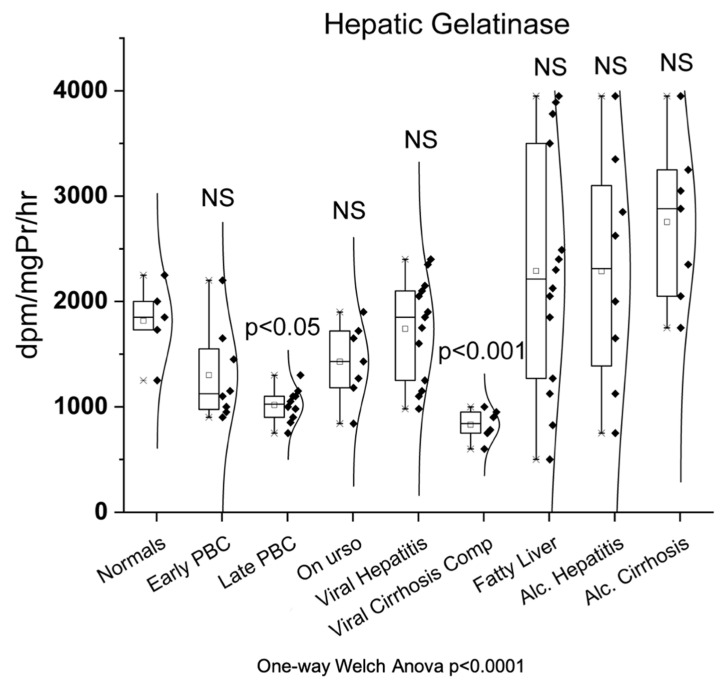
Enzymatic activity of hepatic gelatinase in all studied groups.

**Figure 5 biomedicines-10-03179-f005:**
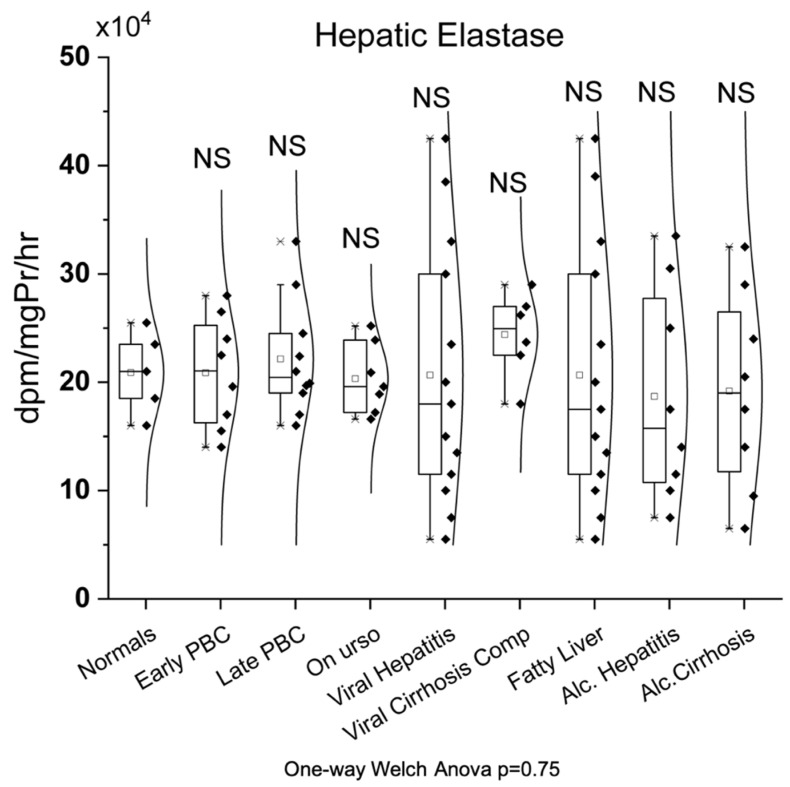
Enzymatic activity of hepatic elastase in all studied groups.

**Figure 6 biomedicines-10-03179-f006:**
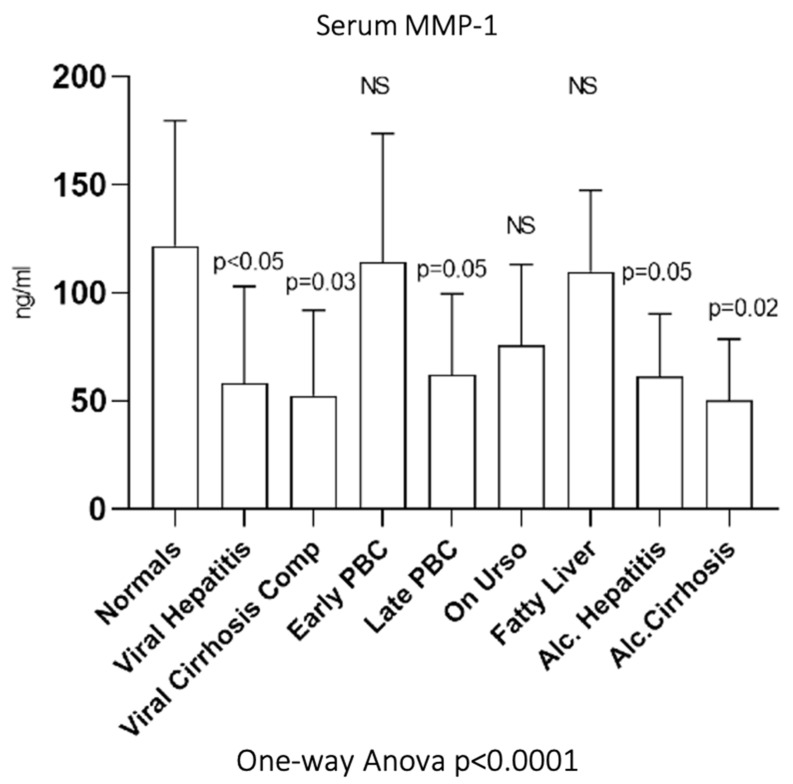
Serum MMP-1 in all studied groups. Mean and standard deviation are presented.

**Figure 7 biomedicines-10-03179-f007:**
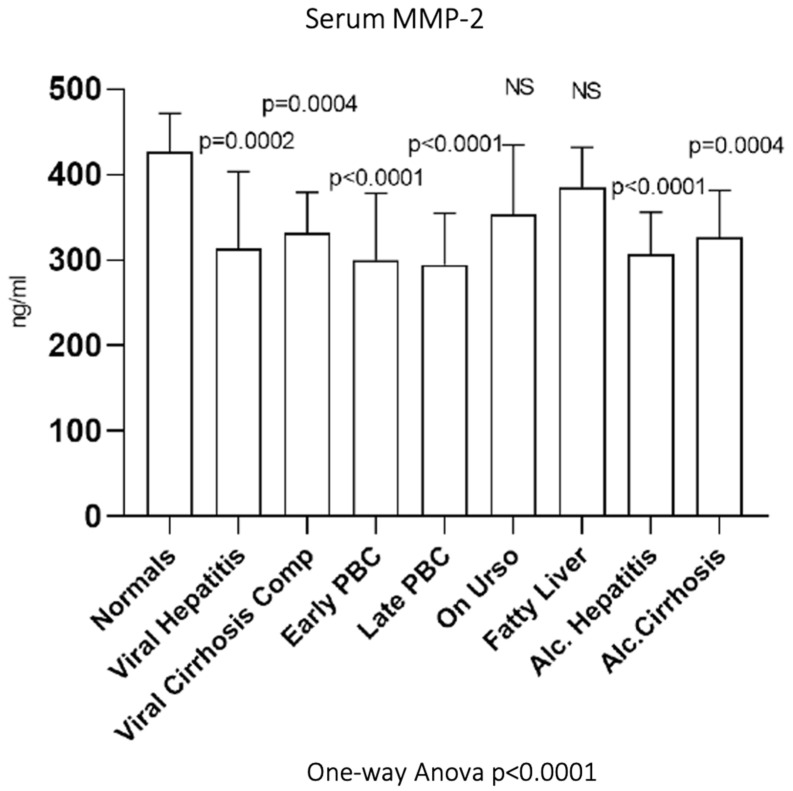
Serum MMP-2 in all studied groups.

**Figure 8 biomedicines-10-03179-f008:**
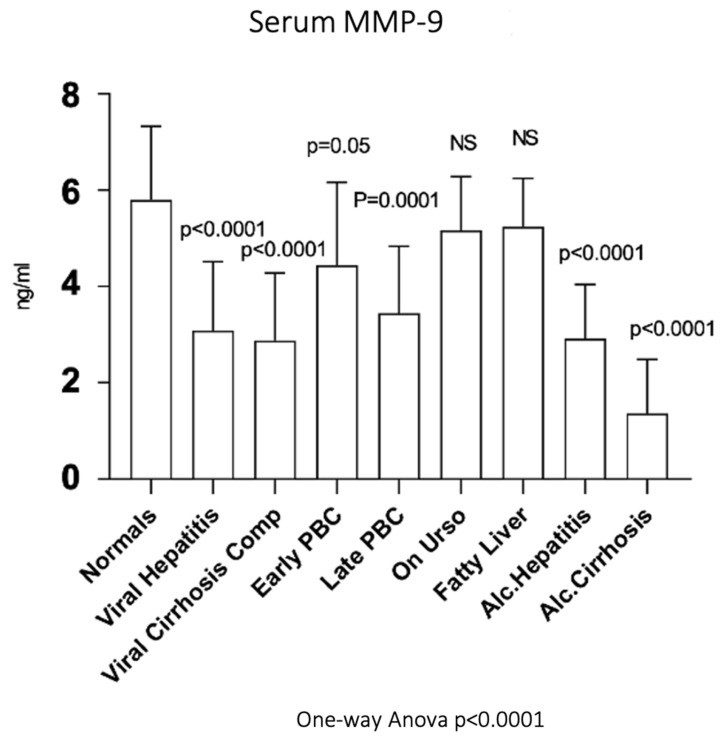
Serum MMP-9 in all studied groups.

**Figure 9 biomedicines-10-03179-f009:**
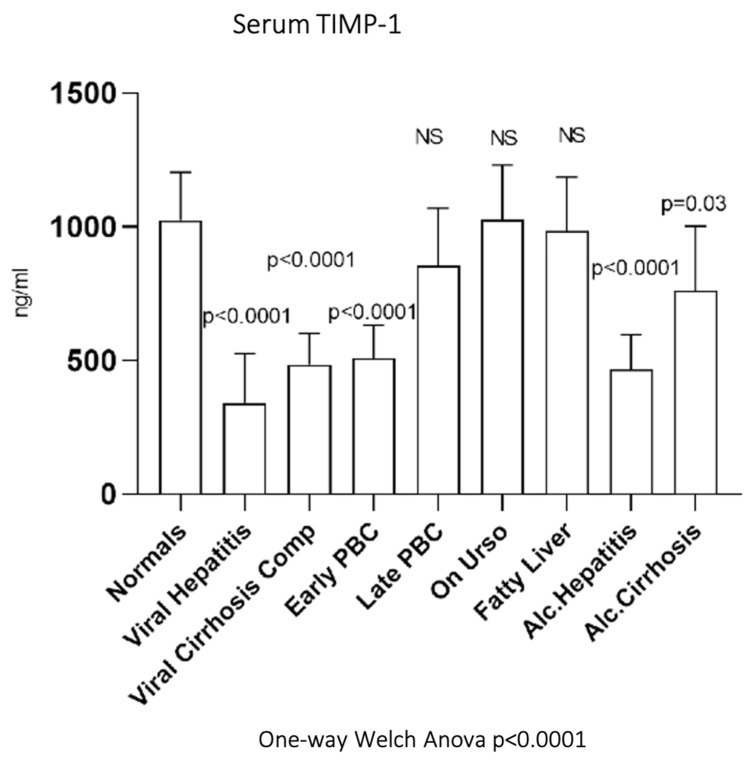
Serum TIMP-1 in all studied groups.

**Figure 10 biomedicines-10-03179-f010:**
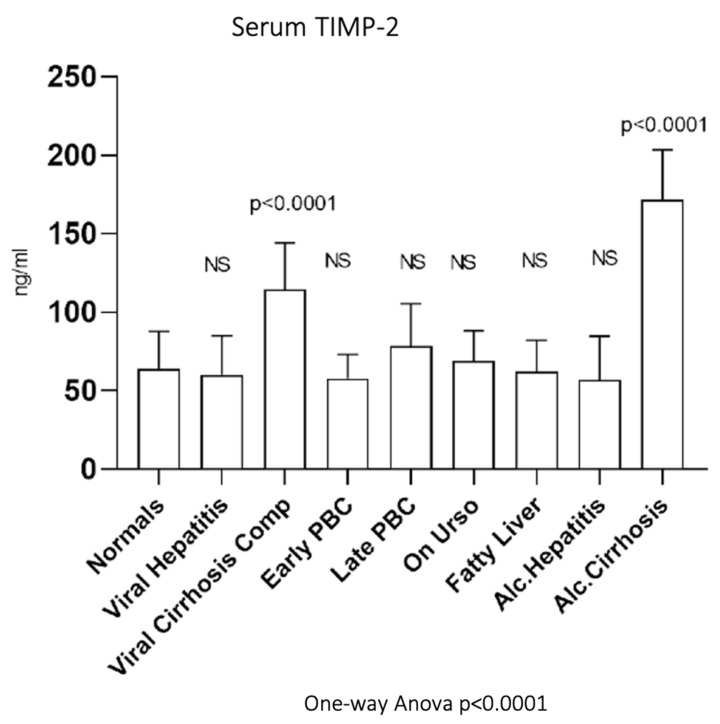
Serum TIMP-2 in all studied groups.

**Table 1 biomedicines-10-03179-t001:** Summary of the enzymatic activity by studied group.

	Liver PH	Liver Collagenase	PH/Collagenase Ratio	Liver Gelatinase	Liver Elastase	MMP1	MMP2	MMP9	TIMP1	TIMP2
Normals	-	-	-	-	-	-	-	-	-	-
Early PBC	▲	NS	NS	NS	NS	NS	▼	▼	▼	NS
Late PBC	▲	▼	▲	▼	NS	▼	▼	▼	NS	NS
On UDCA	NS	NS	NS	NS	NS	NS	NS	NS	NS	NS
Viral hepatitis	NS	▼	▲	NS	NS	▼	▼	▼	▼	NS
Viral cirrhosis	NS	▼	▲	▼	NS	▼	▼	▼	▼	▲
Fatty liver	NS	NS	NS	NS	NS	NS	NS	NS	NS	NS
Alcoholic hepatitis	▲	▼	▲	NS	NS	▼	▼	▼	▼	NS
Alcoholic cirrhosis	▲	▼	▲	NS	NS	▼	▼	▼	▼	▲

Footnotes; PH: prolyl hydroxylase; MMP: matrix metalloproteinases; UDCA: ursodeoxycholic acid; PBC: primary biliary cholangitis; TIMP: tissue inhibitors of metalloproteinases; ▼: statistically significant decrease; ▲: statistically significant increase; NS: non-significant.

## Data Availability

The datasets generated and analyzed during the current study are available from the corresponding author upon reasonable request.

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
