# Peer review of "Enzymes of Fibrosis in Chronic Liver Disease"

_biomedicines, 2022, doi:10.3390/biomedicines10123179_

Round 1

Reviewer 1 Report (Previous Reviewer 1)

Author Response

Reviewer 2 Report (Previous Reviewer 2)

Compared to the previously submitted manuscript, the authors have revised and improved the overall manuscript significantly.  According to the reviewers' suggestions, the abstract and conclusion of the manuscript have been significantly revised. The authors have also performed a statistical analysis and indicated statistical significance. However, there are many ways the authors can improve the quality of the manuscript.

 The Discussion section contains a large amount of published information rather than current experimental findings. It is therefore important that authors discuss their current data in detail with the relevant literature.

  It is necessary to improve the manuscript's language.

Author Response

We thank the reviewers for their comments. All changes are highlighted in yellow (first revision) and green (second revision). Discussion has been modified by omission of paragraphs not directly related to the findings.

Reviewer 2

Comments and Suggestions for Authors

Compared to the previously submitted manuscript, the authors have revised and improved the overall manuscript significantly.  According to the reviewers' suggestions, the abstract and conclusion of the manuscript have been significantly revised. The authors have also performed a statistical analysis and indicated statistical significance. However, there are many ways the authors can improve the quality of the manuscript.

 The Discussion section contains a large amount of published information rather than current experimental findings. It is therefore important that authors discuss their current data in detail with the relevant literature.

We did as you suggested. Information not directly related with our findings were removed from the discussion and our findings were discussed in relation to the relevant literature.

  It is necessary to improve the manuscript's language.

The language was checked by a teacher of English.

Round 2

Reviewer 1 Report (Previous Reviewer 1)

Authors satisfactorily adjusted the manuscript.

This manuscript is a resubmission of an earlier submission. The following is a list of the peer review reports and author responses from that submission.

Round 1

Reviewer 1 Report

Enzymatic pathways of fibrosis in Primary Biliary Cholangitis
Tsomidis et.al.
Summary:
This manuscript evaluates the enzymatic pathways of liver fibrosis in patients (140 total) that were either
normal, early PBC, late PBC, or other liver disease pathology as determined by liver biopsy. The authors
evaluated several enzymatic pathways (PH, MMP, etc.) via standard biochemical analysis. Based on the
enzymatic activity of this pathway, the authors concluded that collagen synthesis is increased in early PBC
while differentially affected throughout other liver disease models including later stage PBC.
Major Issues:
For all figures, all data points should be shown. Violin plots may be more appropriate for Figures 1-5.
There is no discussion of statistical analysis throughout the manuscript. Authors should refer to SAMPL
guidelines for the appropriate reporting of statistics. In the present form, this work cannot be properly
reproduced.
The intent of the study design is unclear what is the relevance of the on urso population? This
does not add any significant conclusions or innovation within the design or conclusions. Are the
authors evaluating PPAR-regulated pathways? Why was urso used rather than more clinically relevant
PPAR ligands?
Of the populations that had significant changes in enzymatic pathways, were differences between sex
evaluated? There is a significant amount of research dedicated to evaluating sexual dipmorphism in
the development and progression of liver diseases. This should be addressed regardless of whether
the authors observed differences.
Since the authors had biopsies to confirm disease progression, histological examples should be
included in the data. Again, this is an issue of rigor and reproducibility.
Serum levels of these enzymes can obviously play a significant role in the progression of PBC and liver
diseases, however, the authors should include at minimum histological evaluation of collagen
deposition (trichrome or Sirius red staining would be sufficient here) and an analysis of rate-limiting
factors involved in the development of fibrosis within these samples.
There is insufficient evidence to support the conclusions in this manuscript.

Reviewer 2 Report

Title: Enzymic pathways of fibrosis in Primary Biliary Cholangitis.

The research data reported in this article has significant novelty and interest which will assist the researchers who are working in this field to gain a better understanding of the enzymatic changes in the formation of fibrosis in primary biliary cholangitis. In order to improve the quality of data presentation and its interpretation, authors need to pay special attention to several things.

 There are a number of information's in the abstract that are not sufficient and clear enough to be easily understood by the reader. Therefore, authors are advised to rewrite the whole abstract in order to include all the necessary information.

Very few data exist on the enzymatic final stage of collagen synthesis (prolyl hydroxylase, PH) and degradation (MMPs). Degradation abbreviation (MMPs): what does it mean?

 In a sequential order, authors should include important information about Primary Biliary Cholangitis (PBC) such as PBC status worldwide, strategy and management of PBC, the reasone mechanisms of PBC, and other necessary information. Presently, the included information’s are not relevant and sufficient to consider a scientific article.

The basic driver of fibrosis is the damage hepatocytes and the resultant sterile inflammation. What does sterile inflammation mean?

In this study, it would be appreciated if the authors could also provide information regarding the presence of other markers related to PBC in the patients.

 There is a need for authors to concentrate their interpretation of data by referring to relevant literature and

 The conclusion of the current article is not sufficient. Therefore, the authors should improve it with suggestions.

 There is a need to improve the language of the manuscript.